# Dietary Thymol Supplementation Promotes Antioxidant Responses and Thermal Stress Resistance in Rainbow Trout, *Oncorhynchus mykiss*

**DOI:** 10.3390/ani14202988

**Published:** 2024-10-16

**Authors:** Morteza Yousefi, Seyyed Morteza Hoseini, Yury Anatolyevich Vatnikov, Arfenya Karamyan, Evgeny Vladimirovich Kulikov

**Affiliations:** 1Department of Veterinary Medicine, RUDN University, 6 Miklukho-Maklaya St, 117198 Moscow, Russia; vatnikov-ya@rudn.ru (Y.A.V.); karamyan-as@rudn.ru (A.K.); kulikov-ev@rudn.ru (E.V.K.); 2Inland Waters Aquatics Resources Research Center, Iranian Fisheries Sciences Research Institute, Agricultural Research, Education and Extension Organization, Gorgan P.O. Box 14965-149, Iran

**Keywords:** aquaculture, functional feed, immunology, nano-liposome, nutrition

## Abstract

Global warming is an important issue in the context of climate change affecting cold-water aquaculture. Heat stress induces oxidative stress in fish, hence the use of a functional diet supplemented with antioxidant additives, such as thymol, can be useful to counteract the impact of global warming on aquaculture. In this study, we used thymol-enriched diets to rear rainbow trout fingerlings over an 8-week period followed by a period of heat stress of 48 h. Although thymol had no significant benefits on the growth and innate immunity of the fish, it improved antioxidant capacity and reduced lipid peroxidation and mortality after heat stress. The appropriate thymol dose for these benefits is 100 mg/kg.

## 1. Introduction

Rainbow trout, *Oncorhynchus mykiss*, is a potential aquaculture species where cold water is available. This species is characterized by high growth rate and feed efficiency and tolerating intensive culture. The annual production of rainbow trout in the world was over 952,000 metric tons in 2021, according to the Food and Agriculture Organization; however, the rainbow trout culture industry faces various obstacles, similarly to other production industries [1]. A global issue for aquaculture industry is climate change and warming, particularly in the cold-water aquaculture sector [2]. Increases in the average temperature of the Earth results in thermal stress in fish, because they are poikilothermic creatures. Thermal stress broadly affects physiological processes, including oxidative stress physiology in fishes, for example; it triggers the stress axis, leading to higher energy expenditure and immunosuppression [3,4]. This is because temperature can influence mitochodrial respiration and associated events to produce high reactive oxygen species (ROS) in aquatic animals [5]. Therefore, having a proper antioxidant function helps the fish to resist heat stresses [6]. As a result, using functional feeds supplemented with antioxidant additives can be helpful for fish under heat stress.

One of the well-known classes of feed additives are herbal supplements [7], which can help fish tolerate heat stress. For example, oregano essential oil in Nile tilapia, *Oreochromis niloticus* [8]; *Laminaria* sp. meal in Atlantic salmon, *Salmo salar* [9]; and Hyssop, *Hyssopus officinalis*, methanolic extract in rainbow trout [6] improved heat stress tolerance in fish. Nature-identical compounds are synthetic form of herbal bioactive compounds. They are readily available with constant purity; thus, they have received significant attention in aquafeed industry [10]. Thymol is a natural monoterpene found in various plants, particularly thyme [11]. It is a well-known antioxidant compound that has been studied in some aquaculture species. For example, snakehead, *Channa argus*, showed improvements in growth rate, feed efficiency and immune/antioxidant parameters, when fed diets containing 300–600 mg/kg thymol; but the highest resistance against a bacterial infection was achieved at a 750 mg/kg thymol concentration [12]. Morselli et al. [13] reported an improved growth rate and feed efficiency, antioxidant responses and oxidative status in grass carp, *Ctenopharyngodon idella*, when fed a diet containing 100 mg/kg of thymol (but not higher levels), although the overall performance of the fish in all treatment groups was poor. In Nile tilapia, variable results were obtained when fish were fed diets supplemented with 250–2000 mg/kg of thymol [14,15]. The only study on rainbow trout has been conducted on ~35 g individuals, where, thymol in the range of 1000–2500 mg/kg steadily improved growth, feed efficiency, antioxidant/immune parameters and disease resistance, but the overall performance of the fish was poor in all treatment groups [16]. There is a lack of data regarding the potential effects of dietary thymol on rainbow trout at a lower individual weight as well as in response to thermal stress. So, the present study assessed the effects of dietary thymol on rainbow trout fingerlings (~5 g) under thermal stress.

## 2. Materials and Methods

### 2.1. Diets and Feeding Trial

Thymol (Sigmaaldrich Co., Saint Louis, MO, USA) was added to diets at concentrations of 50, 100, 200, 400 and 800 mg/kg [13]. A control diet without thymol supplementation was also prepared. Feedstuffs were mixed together for 15 min before being pelleted (2 mm in diameter) by a meat grinder. Thymol was finely pulverized and mixed with dietary oils before being added to the feedstuff. Dietary formulation and proximate composition are presented in Table 1.

Rainbow trout (~5 g individual weight) were purchased from a local farm (Sari, Mazandaran province, Iran) and transported to the laboratory and disinfected by 250 µL/L formalin for 30 min. The fish were kept in a 500 L tank for one week and fed the control diet for acclimatization to the experimental conditions. Then, 360 fish (visually healthy) were randomly distributed into eighteen aquaria (45 L) at a density of 20 fish per aquarium (~2.44 g/L) and were fed the above diets (in triplicate) to apparent satiety, twice a day, for eight weeks. Each aquarium was connected to a semi-closed recirculating system in which the water passed through a physical filter (a piece of felt of 2 mm thickness), biological filter (8 L of pumice with a ~3 cm diameter) and a UV disinfecting chamber (UVC-9 W waterproof plug; SOBO Co., Jiaxing, China), once an hour. The felts were washed daily to remove the collected fish wastes, and the wastes at the aquaria bottom were siphoned daily by renewing 30% of total aquaria water. Water temperature, pH and dissolved oxygen were measured daily by a portable water checker (Hach HQD40, Loveland, CO, USA) and were 12.7 ± 0.43 °C, 7.88 ± 0.26 and 7.84 ± 0.84 mg/L (mean ± standard deviation), respectively. Water total ammonia, nitrite and nitrate concentrations were measured bi-weekly using a photometer (Palintest 7100, Gateshead, UK) and were 1.39 ± 0.57 mg/L, 0.08 ± 0.03 mg/L and 11.4 ± 2.14 mg/L (mean ± standard deviation), respectively.

### 2.2. Growth Performance Calculations

Total feed intake and biomass of each aquarium were measured at the end of the feeding trial and used for the calculation of growth performance and feed efficiency as follows: Weight gain (%) = [100 × (final biomass − initial biomass)/initial biomass)] Specific growth rate (SGR; %/d) = 100 × {[ln(final weight) − ln(initial weight)]/[days]}Feed efficiency (%) = 100 × [(final biomass − initial biomass)/feed intake](1)

### 2.3. Thermal Stress

All treatment groups were subjected to thermal stress after eight weeks. For this, ten fish per aquarium were transferred to new aquaria, in which water temperature was increased to 25 °C using submersible heaters, equipped with a thermostat. Water temperature increased from 13 to 25 °C at a rate of 1 °C per hour and the fish were held at 25 °C for further 48 h [17]. The aquaria were checked every 6 h to remove dead fish.

### 2.4. Sample Collection and Processing

Blood and liver samples were collected from all treatment groups before and after the thermal stress. Three fish were caught per aquarium and anesthetized in clove extract (2 g/L). Blood was withdrawn from the fish caudal vein using heparinized syringes. The blood plasma was separated by centrifuging at 5000 rpm at 4 °C for 7 min. The plasma samples were kept at −70 °C until analysis. The packed erythrocytes were washed with NaCl solution (0.85%) followed by suspensions in phosphate buffer (50 mM; pH 7.0). The suspension was quickly frozen in liquid nitrogen and kept at −70 °C until analysis. After blood sampling, the fish were killed through a sharp blow on the head and spinal cord severance. The abdominal cavity was opened using scissors and a piece of fish liver was dissected, washed with cold phosphate buffer (pH 7.0), frozen in liquid nitrogen and kept at −70 °C until analysis.

### 2.5. Analytical Procedures

#### 2.5.1. Proximate Composition of Diets

Nitrogen (kjeldahl method), ether extract (Soxhlet apparatus), total mineral (burning in furnace) and dry matter (drying in oven) contents of the diets were determined according to AOAC [18].

#### 2.5.2. Plasma Analysis

Plasma total protein, albumin, globulin, total immunoglobulin (Ig), lysozyme and alternative complement (ACH50) were measured at the end of the feeding trial. Total plasma protein (biuret method) and albumin (bromocresol green method) levels were measured using commercial kits (Zist Chem Co., Tehran, Iran). Plasma globulin was calculated by subtracting total protein and albumin levels.

Total plasma Ig levels were measured according to a previous study [19]. Briefly, equal volumes of plasma and polyethylene glycol (12%) were mixed and shaken for 2 h, followed by 15 min centrifuging (5000 rpm) to precipitate Ig. Differences in total protein levels before and after precipitation were equivalent to total Ig.

Plasma lysozyme activity was measured based on bacterial lysis [20]. Briefly, *Micrococcus luteus* was suspended in phosphate buffer (pH 6.2) to obtain an optical density of 0.500–0.600 at 500 nm. Plasma (30 µL) was mixed with the suspension and decrease in optical density was recorded for 5 min. Each 0.001 decrease in optical density per min was equal to one unit of lysozyme.

Plasma ACH50 activity was measured according to Yano [21] with some modifications. Briefly, sheep erythrocytes were suspended in a barbital buffer containing gelatin, EGTA and magnesium. Plasma samples were diluted in the same buffer at a ratio of 5:95 (*v*:*v*) and 50 µL of the diluted sample was mixed with an equal volume of the sheep erythrocyte suspension and 50 µL of the buffer. After 90 min incubation at room temperature, the percentage of hemolysis was determined at 412 nm.

Plasma cortisol, glucose, total antioxidant capacity (TAC) and ascorbate concentrations were measured before and after thermal stress. Plasma cortisol was measured based on the enzyme-linked immunosorbent assay using a commercial kit with a detection range of 0–500 ng/mL (Monobind Co., Lake Forest, CA, USA). Inter- and intra-assay variations were 7.12 and 9.13%, respectively. Plasma glucose levels were measured based on the glucose–oxidase method using a commercial kit (Zist Chem Co., Tehran, Iran). Plasma TAC was measured based on the ferric-reducing capacity of the samples using a commercial kit (Zellbio Co., Lonsee, Germany). Plasma ascorbate concentration was determined based on the reaction with 2,4-dinitrophenylhydrazine, which produces a red color [22].

#### 2.5.3. Liver and Erythrocyte Sample Analysis

The liver samples were homogenized in 2 volumes (*w*:*v*) of cold phosphate buffer (100 mM; pH 7.0) and centrifuged (13,000 rpm, 15 min, 4 °C) to obtain enzyme extract. The erythrocyte suspensions were thawed and centrifuged in the same manner. The soluble proteins of the enzyme extracts were determined according to the method of Bradford [23]. Hepatic and erythrocyte superoxide dismutase (SOD) activities were determined based on the inhibition of the autoxidation of pyrogallol for 2 min at 420 nm [24]. Hepatic and erythrocyte catalase (CAT) activities were determined based on the decomposition of hydrogen peroxide within 1 min, following yellow coloration by adding ammonium molybdate measured at 420 nm [25]. Hepatic glutathione peroxidase (GPx) activity was determined based on the consumption of reduced glutathione (GSH) for 15 min and the reaction of the remaining GSH with the Elman reagent at 420 nm [26]. Hepatic glutathione reductase (Gr) activity was determined using a commercial kit (Zellbio Co., Lonsee, Germany) at 340 nm. Hepatic and erythrocyte GSH concentrations were measured based on color development after mixing with the Elman reagent. Hepatic and erythrocyte malondialdehyde (MDA) contents were determined according the reaction with thiobarbituric acid and heating for 60 min (95 °C) and by measuring the red color produced at 535 nm [27].

### 2.6. Statistical Analysis

Based on the quadratic regression and polynomial contrast analysis, no significant and reliable models were found between the thymol levels in the diets and tested parameters. Therefore, the data were subjected to ANOVA to find the significant effects of dietary thymol concentrations on the tested parameters. Growth and plasma immunological parameters and post-stress survival data were analyzed by one-way ANOVA. Plasma stress markers and hepatic/erythrocyte antioxidant parameters were analyzed by repeated measure two-way ANOVA to find significant interactions between dietary thymol and thermal stress. Multiple comparisons among the treatment groups were conducted using Duncan’s test. α was set at 0.05, and all analyzes were performed in SPSS v.22.

## 3. Results

### 3.1. Growth Performance

There was no mortality in the treatment groups during the rearing period. Dietary thymol levels did not significantly affect the growth performances of the fish or the feed efficiency (Table 2).

### 3.2. Humoral Immunological Parameters

Dietary thymol levels had no significant effects on plasma immunological parameters, including lysozyme, ACH50, total Ig, total protein, albumin and globulin levels (Table 3).

### 3.3. Post-Stress Survival

Fish survival after the thermal stress showed significant differences among the treatment groups, where significant elevations were found in survival of the fish fed diets containing 50–400 mg/kg of thymol (Figure 1).

### 3.4. Erythrocyte Antioxidant Parameters

Thymol had no significant effects on erythrocyte CAT and GSH. Thermal stress resulted in significant elevations in erythrocyte CAT activity, but GSH level decreased. Thymol and thermal stress had interaction effects on erythrocyte SOD activity and MDA level. Before heat stress, significant elevation in erythrocyte SOD activity was observed in 100 mg/kg thymol treatment groups; however, there were no significant differences in the enzyme activities among the treatment groups after thermal stress. There were no significant differences in erythrocyte MDA levels among the treatment groups before stress. Thermal stress significantly increased erythrocyte MDA level in all treatment groups, except in the 100 and 200 mg/kg thymol treatment groups (Table 4).

### 3.5. Hepatic Antioxidant Parameters

Dietary thymol showed no significant effects on plasma cortisol and glucose concentrations, but both parameters showed significant elevations after thermal stress. There were no interactions between dietary thymol and thermal stress on these parameters (Figure 2).

Plasma TAC and ascorbate levels showed significant changes in response to dietary thymol concentrations (Figure 3). Fish fed diets containing 100 and 200 mg/kg of thymol had similar plasma TAC and ascorbate levels that were significantly higher than the other treatment groups. Plasma TAC was similar among the 0, 50, 400 and 800 mg/kg thymol treatment groups. Plasma ascorbate levels in the 50, 100 and 200 mg/kg thymol treatment groups were significantly higher than that of 0 mg/kg thymol treatment groups. The highest plasma ascorbate level was seen in the 100 mg/kg thymol treatment group. Thermal stress significantly decreased plasma TAC and ascorbate levels, but there were no interactions between the thermal stress and dietary thymol levels for these parameters.

There were interactions between dietary thymol levels and thermal stress on hepatic SOD and CAT activities (Figure 4). Before heat stress, hepatic SOD activity significantly increased in 100 mg/kg thymol treatment group compared to 0 mg/kg treatment group. Thermal stress led to a significant increase in hepatic SOD activity in the 0, 50, 400 and 800 mg/kg thymol treatment groups and the highest activity was seen in the 400 and 800 mg/kg treatment groups. There was no significant change in hepatic SOD activity of 100 and 200 mg/kg thymol treatment groups, before and after thermal stress. There were no significant differences in hepatic CAT activities among the treatment groups before heat stress. After the thermal stress, hepatic CAT activity significantly increased in all treatment groups, except the 100 mg/kg thymol group. Hepatic GPx activity significantly increased in fish fed the 200 mg/kg thymol diet, compared to those fed the 0 mg/kg diet. Hepatic Gr activity significantly increased in fish fed the 50–200 mg/kg thymol diet, compared to those fed the 0 mg/kg diet, and the highest activity was found in the 100 mg/kg treatment group. Thermal stress significantly increased hepatic GPx and Gr activities; however, there were no interactions between thermal stress and dietary thymol levels (Figure 4).

Hepatic GSH contents significantly increased in 100–400 mg/kg of thymol treatment groups, compared to 0 mg/kg, and the highest content was related to 100 mg/kg. Thermal stress significantly decreased hepatic GSH levels, but there were no interactions between thermal stress and dietary thymol levels (Figure 5). There was an interaction between thermal stress and dietary thymol on hepatic MDA concentrations. Before heat stress, fish fed a 100 mg/kg thymol diet had significantly lower hepatic MDA content compared to those fed a 0 mg/kg diet. Thermal stress significantly increased hepatic MDA content in the 0, 50, 400, and 800 mg/kg thymol treatment groups, but not in the 100 and 200 mg/kg groups. The highest hepatic MDA concentration after heat stress was observed in 0 and 800 mg/kg thymol treatment groups (Figure 5).

## 4. Discussion

This is the first study aiming to determine the optimum dietary thymol level for rainbow trout fingerlings. According to the present results, it may be stated that thymol is not a growth-promoting agent in rainbow trout fingerlings. Against the present findings, growth-promoting effects of dietary thymol have been previously reported in rainbow trout with an initial fish weight of ~35 g, where thymol in a range of 1000–2500 mg/kg steadily improved growth and feed efficiency [16]. The exact reasons for this controversy are not known, but these concentrations seem high compared to those reported in other fish species such as snakehead (300 mg/kg [12]) and grass carp (100 mg/kg [13]). Furthermore, the growth performance and feed efficiency of rainbow trout in the study conducted by Hafsan, Saleh, Zabibah, Obaid, Jabbar, Mustafa, Sultan, Gabr, Ramírez-Coronel, Khodadadi and Dadras [16] were inferior, making it doubtful that the fish showed their highest growth potentials under the study conditions. On the other hand, the present findings are in line with those found in a study of Nile tilapia fed diets supplemented with 250 and 500 mg/kg thymol [14].

Thymol showed no benefits on humoral immune parameters in the present study, which is not in line with previous studies on snakehead [12] and 35 g rainbow trout [16]. It has been found that thymol can improve plasma lysozyme, total protein, complement and Ig in 35 g rainbow trout [16]; although an inverse response of bactericidal activity and poor growth performance of the fish make it doubtful whether thymol had immunostimulant effects. Further studies should focus on immune responses in other tissues (e.g., skin mucus, intestines and head kidney) of rainbow trout fingerlings to find if thymol can improve them.

Increases in water temperature trigger thermal stress axis in fish characterized by elevations in blood cortisol and glucose [6]. These changes provide the required energy to cope with the negative consequences of the thermal stress [28]. The present results are in line with LeBlanc et al. [29] and Yousefi, Hoseini, Kulikov, Seleznev, Petrov, Babichev, Kochneva and Davies [6] who found increases in blood cortisol and glucose levels in rainbow trout after thermal stress. However, thymol showed no anti-stress effects in the fish during the thermal stress. There is no similar study for comparison with the present results; although other herbal supplements such as lion’s mane meal [30] and Hyssop methanolic extract [6] mitigated stress responses to water temperature elevation.

Fish liver is one of the most metabolically active organs that participates in antioxidant and detoxification processes. Fish show the activation of the hepatic antioxidant system and the consumption of cellular antioxidant compounds when exposed to high water temperatures [31]. This may be due to acceleration in basal metabolism and a higher energy expenditure under heat stress conditions, with both leading to increases in catabolic processes and the respiration and formation of pro-oxidant compounds like superoxide ions [32]. Ascorbate is one of the reducing compounds in living cells that is oxidized by dehydroascorbate to neutralize pro-oxidants [33]. Also, GSH has radical scavenging activity and serves as a co-factor in glutathione-dependent antioxidant enzymes to protect cells against pro-oxidants [34]. We found significant declines in hepatic ascorbate and GSH after the thermal stress. Also, the total reducing capacity of the fish liver decreased after thermal stress; all these changes collectively suggest that the rate of pro-oxidant neutralization increased after heat stress. Similarly, the antioxidant enzymes showed elevations in activity; however, an increase in the hepatic MDA concentration suggests there were some levels of oxidative stress in the fish liver after the thermal stress. These results are in line with previous studies on Wuchang bream, *Megalobrama amblycephala* [35], and rainbow trout [6] exposed to thermal stress.

Besides liver, erythrocytes have a critical dependency on antioxidants for proper functioning due to the presence of the heme protein and constant exposure to oxygen molecules [36]. This is because of the presence of the heme protein and constant exposure to oxygen molecules [37,38]. The present study showed that thermal stress led to the activation of SOD and CAT activity and the depletion of GSH and the occurrence of lipid peroxidation in the fish erythrocyte. These changes are in line with those found in *Notothenia coriiceps* and *Notothenia rossii* subjected to high water temperature [38,39].

Thymol is a powerful antioxidant and is known to be beneficial for fish under certain conditions that trigger oxidative stress [11]. For example, dietary thymol supplementation significantly improved the activities of SOD, CAT and GSH reserves in different tissues of Nile tilapia and mitigated lipid peroxidation and protein carbonylation after exposure to zinc nanoparticles [15]. Moreover, the pre-treatments of rainbow trout and snakehead with thymol-supplemented diets resulted in improvements in antioxidant power and subsequent resistance against bacterial infections [12,16]. As infections induce severe oxidative stress, the higher fish resistance may be attributed to the higher antioxidant power. The present study showed for the first time that dietary thymol can increase the resistance of fish to thermal stress and the results suggest that this benefit related to the improvement in antioxidant power in the fish liver and erythrocyte. Although the responses were tissue-specific, thymol clearly suppressed lipid peroxidation by improving antioxidant enzymes in the liver and erythrocyte of rainbow trout. Erythrocyte health is pivotal during thermal stress, as the need for gas mobilization increases under this condition [40]. Moreover, a well-functioning antioxidant system in the fish liver protects the fish body against excess pro-oxidant formation during heat stress.

## 5. Conclusions

The present study showed that dietary thymol has no benefits on growth performance and humoral innate immune responses in rainbow trout fingerlings. Nevertheless, thymol is capable of increasing fish resistance to acute thermal stress, mainly by improving the enzymatic/non-enzymatic antioxidant responses of the fishes’ erythrocytes and livers. According to the results, 100 mg/kg of thymol is the best concentration to achieve the above-mentioned benefits.

## Figures and Tables

**Figure 1 animals-14-02988-f001:**
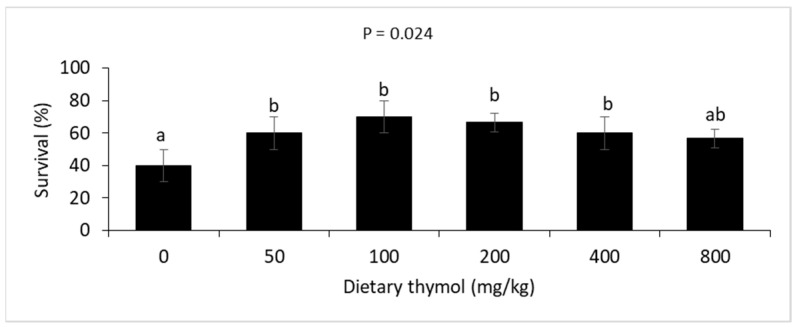
Survival (mean ± standard deviation) of rainbow trout fed 0–800 mg/kg thymol diets over eight weeks and subjected to 48 h of thermal stress. Different letters above the bars show significant differences among the thymol treatment groups (*n* = 3; Duncan’s test).

**Figure 2 animals-14-02988-f002:**
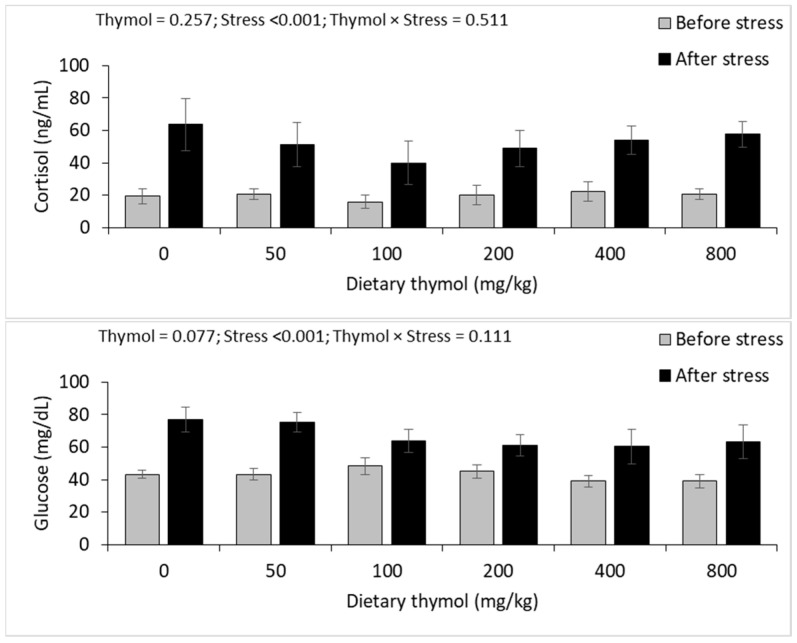
Plasma cortisol and glucose (mean ± standard deviation) of rainbow trout fed 0–800 mg/kg thymol diets over eight weeks and subjected to a 48 h thermal stress (*n* = 3; Duncan test).

**Figure 3 animals-14-02988-f003:**
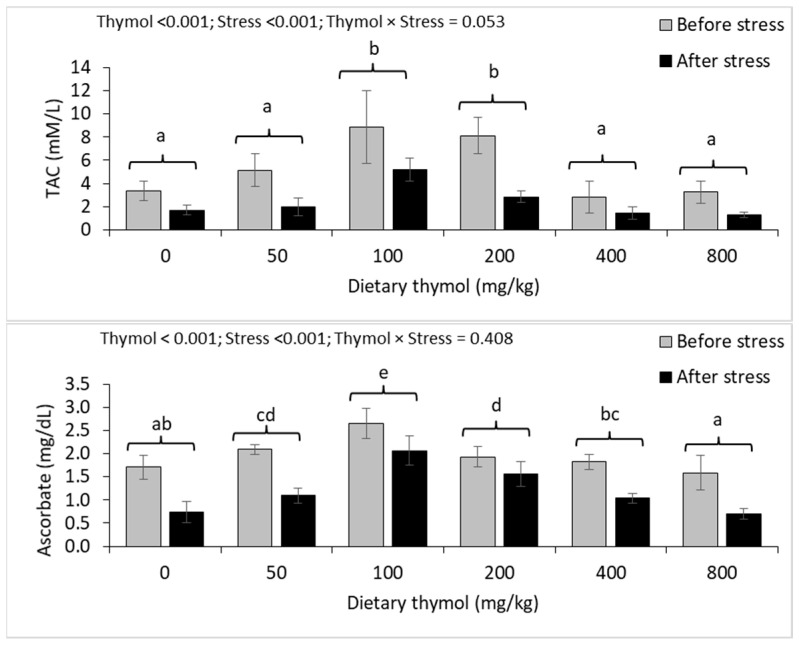
Plasma TAC and ascorbate levels (mean ± standard deviation) of rainbow trout fed 0–800 mg/kg thymol diets over eight weeks and subjected to a 48 h thermal stress. Different letters above the bars show significant differences among the thymol treatment groups (*n* = 3; Duncan’s test).

**Figure 4 animals-14-02988-f004:**
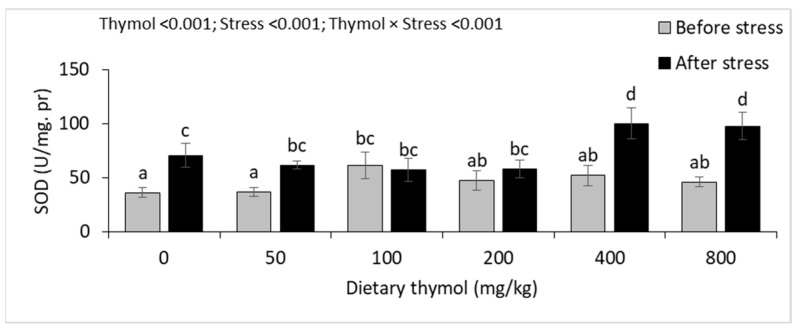
Hepatic activity of the antioxidant enzymes (mean ± standard deviation) of rainbow trout fed 0–800 mg/kg thymol diets over eight weeks and subjected to 48 h of thermal stress. Different letters above the bars show significant differences among the thymol treatment groups (*n* = 3; Duncan’s test).

**Figure 5 animals-14-02988-f005:**
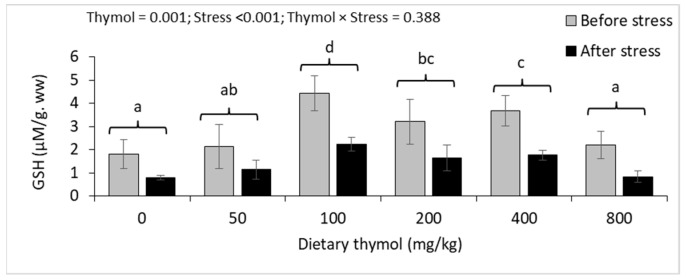
Hepatic GSH and MDA concentrations (mean ± standard deviation) of rainbow trout fed 0–800 mg/kg thymol diets over eight weeks and subjected to 48 h of thermal stress. Different letters above the bars show significant differences among the thymol treatment groups (*n* = 3; Duncan’s test).

**Table 1 animals-14-02988-t001:** Composition of the diets (the amounts are given in g/kg).

	CTL	50 TM	100 TM	200 TM	400 TM	800 TM
Corn meal	50	50	50	50	50	50
Wheat meal	220	220	220	220	220	220
Soybean meal	170	170	170	170	170	170
Soybean oil	70	70	70	70	70	70
Kilka fishmeal ^1^	254	254	254	254	254	254
Fish process byproduct ^2^	130	130	130	130	130	130
Poultry byproduct ^3^	9	9	9	9	9	9
Vitamin premix ^4^	5	5	5	5	5	5
Mineral premix ^4^	5	5	5	5	5	5
Methionine	3	3	3	3	3	3
Lysine	3	3	3	3	3	3
Proximate composition						
Moisture	91.5	90.8	91.6	90.3	91.0	91.3
Crude protein	431.7	435.2	430.6	433.6	432.0	436.9
Crude fat	154.8	152.9	150.4	153.6	155.7	158.3
Crude ash	75.8	76.1	76.3	75.8	76.7	76.3
Crude fiber	28.7	27.6	28.0	28.6	28.7	28.3

^1^ crude protein, 73%; crude fat, 12%; ^2^ crude protein, 56%; crude fat, 15%; ^3^ crude protein, 59%; crude fat, 23%; ^4^ Amineh Gostar Co. (Tehran, Iran).

**Table 2 animals-14-02988-t002:** Growth performance, feed efficiency and survival (mean ± standard deviation) of rainbow trout fed 0–800 mg/kg thymol diets over eight weeks (*n* = 3).

	Dietary Thymol (mg/kg)						
	0	50	100	200	400	800	Sig.
Initial weight (g)	5.53 ± 0.15	5.60 ± 0.10	5.50 ± 0.10	5.50 ± 0.10	5.53 ± 0.21	5.53 ± 0.15	0.956
Final weight (g)	24.7 ± 0.46	25.7 ± 0.67	26.2 ± 1.32	24.2 ± 0.34	25.5 ± 1.33	24.3 ± 1.27	0.152
Weight gain (%)	346 ± 20.1	359 ± 18.8	376 ± 16.9	341 ± 13.6	360 ± 14.9	338 ± 10.9	0.105
SGR (%/d)	2.67 ± 0.08	2.72 ± 0.07	2.78 ± 0.06	2.65 ± 0.05	2.73 ± 0.06	2.63 ± 0.04	0.109
Feed intake (g/tank)	432 ± 8.88	451 ± 15.8	466 ± 24.4	425 ± 8.55	438 ± 25.3	421 ± 21.5	0.088
Feed efficiency (%)	88.6 ± 1.02	89.1 ± 0.64	88.7 ± 1.57	88.2 ± 1.46	91.0 ± 1.62	89.0 ± 1.35	0.221
Survival (%)	100	100	100	100	100	100	-

**Table 3 animals-14-02988-t003:** Humoral immune responses (mean ± standard deviation) of rainbow trout fed 0–800 mg/kg thymol diets over eight weeks (*n* = 3).

	Dietary Thymol (mg/kg)						
	0	50	100	200	400	800	Sig.
Lysozyme (U/mL)	28.0 ± 3.61	28.7 ± 4.04	32.7 ± 6.65	28.0 ± 2.65	26.3 ± 2.52	26.3 ± 3.05	0.452
ACH50 (%)	33.7 ± 6.56	36.0 ± 7.54	41.3 ± 6.11	35.7 ± 5.13	36.3 ± 5.51	37.3 ± 4.72	0.742
Total Ig (g/dL)	6.20 ± 0.95	7.60 ± 1.21	7.53 ± 1.50	6.43 ± 0.84	6.93 ± 1.31	6.27 ± 0.74	0.492
Total protein (g/dL)	3.40 ± 0.40	3.37 ± 0.45	3.43 ± 0.38	3.37 ± 0.31	3.27 ± 0.31	3.20 ± 0.26	0.964
Albumin (g/dL)	1.61 ± 0.18	1.66 ± 0.18	1.65 ± 0.13	1.64 ± 0.12	1.62 ± 0.10	1.60 ± 0.10	0.995
Globulin (g/dL)	1.79 ± 0.25	1.71 ± 0.27	1.79 ± 0.26	1.73 ± 0.18	1.64 ± 0.21	1.60 ± 0.17	0.883

**Table 4 animals-14-02988-t004:** Antioxidant responses in erythrocytes (mean ± standard deviation) of rainbow trout fed 0–800 mg/kg thymol diets over eight weeks and subjected to 48 h of thermal stress. Different superscript letters within a row show significant differences among the thymol treatment groups before (lowercase letters) and after (uppercase letters) thermal stress. Asterisks show significant differences compared to the corresponding values before heat stress (*n* = 3; Duncan’z test).

	Thermal	Dietary Thymol (mg/kg)								
	Stress	0	50	100	200	400	800	Thymol	Stress	Thymol × Stress
SOD (U/mg. pr)	Before	2.28 ± 0.95 ^ab^	2.67 ± 0.71 ^ab^	5.23 ± 1.00 ^c^	3.70 ± 0.80 ^b^	2.79 ± 1.02 ^ab^	1.97 ± 0.40 ^a^	0.016	0.007	0.036
	After	3.39 ± 1.02 ^A^	4.58 ± 0.89 ^A*^	3.88 ± 0.46 ^A^	4.02 ± 0.41 ^A^	4.11 ± 0.58 ^A^	4.18 ± 0.71 ^A*^			
CAT (U/mg. pr)	Before	365 ± 153	405 ± 37.4	398 ± 88.5	420 ± 45.6	400 ± 43.2	419 ± 11.5	0.905	<0.001	0.952
	After	824 ± 128	954 ± 107	919 ± 146	925 ± 178	894 ± 287	975 ± 99.4			
GSH (mM/mg. pr)	Before	2.35 ± 0.77	2.16 ± 0.44	1.85 ± 0.49	1.97 ± 0.10	1.96 ± 0.42	1.99 ± 0.52	0.867	<0.001	0.711
	After	1.21 ± 0.31	1.07 ± 0.22	1.37 ± 0.25	1.32 ± 0.28	1.13 ± 0.31	1.36 ± 0.26			
MDA (nM/mg. pr)	Before	0.10 ± 0.02 ^a^	0.10 ± 0.01 ^a^	0.11 ± 0.02 ^a^	0.10 ± 0.01 ^a^	0.09 ± 0.01 ^a^	0.10 ± 0.01 ^a^	0.013	<0.001	<0.001
	After	0.15 ± 0.02 ^B*^	0.16 ± 0.01 ^B*^	0.09 ± 0.02 ^A^	0.07 ± 0.02 ^A^	0.15 ± 0.01 ^B*^	0.15 ± 0.01 ^B*^			

## Data Availability

Data are available as Appendix A.

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
