# Peer review of "Dietary Thymol Supplementation Promotes Antioxidant Responses and Thermal Stress Resistance in Rainbow Trout, *Oncorhynchus mykiss"

_animals, 2024, doi:10.3390/ani14202988_

Round 1

Reviewer 1 Report

Comments and Suggestions for Authors

The MS provides new data on an environmentally friendly feed additive, thymol, for the enhancement of stress resistance of rainbow trout fingerlings in aquaculture. The Introduction gives the reader a good understanding of the background. The M&M section is sufficient, and results are adequately presented. The Discussion section brings sufficient perspective to the studied issue. However, some minor issues should be addressed throughout the text.

L51 I think it is better to say that excess of pro-oxidant molecules triggers oxidative stress, not the antioxidant system (as antioxidants are always active even at physiological levels of ROS in the cells).

Table 1 - Please add the units.

L102-103. As I understand, “in triplicate” here means the number of experiment replications. Please clarify the number of meals per day.

L168 I believe “Deutschland” is unnecessary here.

L311 Would it not look better to put Yousefi et al. as you did before, e.g., LeBlanc et al. [28]? The same goes for L295.

L340 Please rephrase “its benefits have been approved in fish”, the meaning is not entirely clear.

L354 Mind the reference number.

Comments on the Quality of English Language

I believe there are some minor issues with the quality of English. I recommend showing your manuscript to a qualified colleague for editing.

Author Response

The MS provides new data on an environmentally friendly feed additive, thymol, for the enhancement of stress resistance of rainbow trout fingerlings in aquaculture. The Introduction gives the reader a good understanding of the background. The M&M section is sufficient, and results are adequately presented. The Discussion section brings sufficient perspective to the studied issue. However, some minor issues should be addressed throughout the text.

Response: Thank you very much for the time you spent to read this manuscript.

L51 I think it is better to say that excess of pro-oxidant molecules triggers oxidative stress, not the antioxidant system (as antioxidants are always active even at physiological levels of ROS in the cells).

Response: Thank you. We changed it to “Thermal stress broadly affects physiological processes including oxidative stress phys-iology in fishes for example, it triggers the stress axis leading to higher energy ex-penditure and immunosuppression [3, 4]. It is because temperature can influence mi-tochodrial respiration and associated events to produce high reactive oxygen species (ROS) in aquatic animals [5]. Therefore, having a proper antioxidant function helps the fish to resist heat stresses [6].”.

Table 1 - Please add the units.

Response: Thank you. It was added (g/kg).

L102-103. As I understand, “in triplicate” here means the number of experiment replications. Please clarify the number of meals per day.

Response: Thank you. We fed the fish twice a day. It was added to the text.

L168 I believe “Deutschland” is unnecessary here.

Response: Thank you. It was deleted.

L311 Would it not look better to put Yousefi et al. as you did before, e.g., LeBlanc et al. [28]? The same goes for L295.

Response: Thank you. Please note that we used Endnote to format the references based on the Chicago style, as the journal stated. If it is not correct, we will communicate with the publisher during the production processes.

L340 Please rephrase “its benefits have been approved in fish”, the meaning is not entirely clear.

Responses: Thank you. It was revised to read “Thymol is a powerful antioxidant and have been beneficial in fish under certain conditions that trigger oxidative stress”

L354 Mind the reference number.

Reference: Thank you. It was revised.

Reviewer 2 Report

Comments and Suggestions for Authors

Reviewer’s comments to Authors

The study" Dietary thymol supplement promotes antioxidant responses and thermal stress resistance in rainbow trout, Oncorynchus mykiss" sheds new light on the potential applications of thymol as a dietary supplement in aquaculture production. The use of thymol (a natural bioactive substance) is a promising strategy to improve fish performance and health, which is mostly important in the perspective of viable aquaculture practices. The study is timely, seeing the growing demand for options to synthetic additives in aquafeed that can improve fish growth and protection capacity against ecological stressors. Overall, the paper is well written, and the research question is apt and significant. Nevertheless, there are a few parts that could be improved to enhance the overall impact of the study. For instance,

1. Authors should include a Graphical Abstract (GA)to boost the article's online visibility and appeal upon publication.

2.Lines 42 & 44: citations are required, thereafter be listed in References section

3. Line 70: add …"that"…

4. Table 1 should show all the diets compositions used in this study

5 Line 106: Author should replace a with "an"

Results

6. lines 197-284: Authors should consider using subheadings in the Results section to separate different parameters to improve readability and comprehension. The present format lacks clarity, making it challenging to follow. Organizing results under specific subheadings will help deliver findings more effectively.

7. line 354: [3939]?

8. Authors should follow Journal Reference list style.

Comments on the Quality of English Language

Minor English editing required.

Author Response

The study" Dietary thymol supplement promotes antioxidant responses and thermal stress resistance in rainbow trout, Oncorynchus mykiss" sheds new light on the potential applications of thymol as a dietary supplement in aquaculture production. The use of thymol (a natural bioactive substance) is a promising strategy to improve fish performance and health, which is mostly important in the perspective of viable aquaculture practices. The study is timely, seeing the growing demand for options to synthetic additives in aquafeed that can improve fish growth and protection capacity against ecological stressors. Overall, the paper is well written, and the research question is apt and significant. Nevertheless, there are a few parts that could be improved to enhance the overall impact of the study. For instance,

  1. Authors should include a Graphical Abstract (GA)to boost the article's online visibility and appeal upon publication.

Response: Thank you. We added a GA to the manuscript.

2.Lines 42 & 44: citations are required, thereafter be listed in References section

Response: Thank you. A reference from FAO was added.

  1. Line 70: add …"that"…

Response: Thank you. We did it.

  1. Table 1 should show all the diets compositions used in this study

Response: Thank you. We did it as you suggested.

5 Line 106: Author should replace a with "an"

Response: Thank you. It was revised.

Results

  1. lines 197-284: Authors should consider using subheadings in the Results section to separate different parameters to improve readability and comprehension. The present format lacks clarity, making it challenging to follow. Organizing results under specific subheadings will help deliver findings more effectively.

Response: Thank you. Subheadings were added.

  1. line 354: [3939]?

Response: Thank you. It seems there is a problem in my Endnote software. Anyway, we fixed it in the revised from.

  1. Authors should follow Journal Reference list style.

Response: Thank you. Please note that the journal supports format-free style for reference formatting. So we used the Chicago style is it is similar to the article published by this journal.

Reviewer 3 Report

Comments and Suggestions for Authors

Please make all changes in track change mode. 

The  manuscript animals-323130 describes about the increase in oxidative heath status in rainbow trout with thymol supplementation under thermal stress. Followings are the observations

Simple summary; Please include line what was the background of adding thymol rich diet

Abstract

Line 13-20: what are CT and TM, please remember that abbreviations are mentioned first in full before they are cited in abstract and again in main text.

What  are  0 (CT), 50 19 (50TM), 100 (100TM), 200 (200TM), 400 (400TM) and 800 (800TM), write 50 or 50TM not both. And write like 50, 100, 200, 400 and 800 TM

If above are doses, then mention a line that based on what the doses are prepared.

I wish not to write in details that what is being obtained at what particular dose. Just mention the primary results and discuss in 1-2 lines, state the application and future prospective,

Keywords: Ok

Introduction

Line 40-45: Add references when you present data in a line.

48-49: Change : Thermal stress broadly affects physiological processes including oxidative stress physiology in fishes for example, it triggers the stress axis leading to higher energy expenditure and immunosuppression [].It is because temperature can influence mitochodrial respiration and associated events to produce high reactive oxygen species (ROS) in aquatic animals [https://pubmed.ncbi.nlm.nih.gov/24679979/ and some other refernces}.

Line 50-51: It does not link with the previous line, so change it to “It increases the chance of energy expenditure that leads to produce of excess pro-oxidant molecules………..

Line 54-67: Reduce it to 3-5 lines and add the thymol (next para ie line 69-84) para with this.

79-81: I  suspect about the novelty of the work: If the authors declare that such studied are already there, then what is your current hypothesis?? Please clearly mention the hypothesis based on literature especially reference 15.

Materials and methods

Section 2.1 Based on what you have given this dose. Reference 12 is on craps. You had to determine the doses.

Whether fish were disinfected after collection, if yes, mention how and if no then mention why not??

Section 2.3: Mention why 25 C is thermal stress to this fish.

Section 2.5.3: Mention in detail about the composition of the buffer, its pH, antiproatse etc.

Results

Table 2, 3, figure 1: with  n=3 cant give you a concluding remark on growth rate

Figure 2-5,  need two way ANOVA test

Discussion

OK

Conclusion

Ok

Author Response

The  manuscript animals-323130 describes about the increase in oxidative heath status in rainbow trout with thymol supplementation under thermal stress. Followings are the observations

Simple summary; Please include line what was the background of adding thymol rich diet

Response: Thank you. We added this sentence “Heat stress induces oxidative stress in fish, hence using functional diet supplemented with anti-oxidant additives, such as thymol, can be useful to counteract the impact of global warming on aquaculture.”

Abstract

Line 13-20: what are CT and TM, please remember that abbreviations are mentioned first in full before they are cited in abstract and again in main text.

Response: Thank you. Please note that we described them in the first line of abstract.

What  are  0 (CT), 50 19 (50TM), 100 (100TM), 200 (200TM), 400 (400TM) and 800 (800TM), write 50 or 50TM not both. And write like 50, 100, 200, 400 and 800 TM

Response: Thank you. Please note that, for example, “50TM” refers to a diet contining 50 mg/kg thymol. In fact we used these codes for the diets.

If above are doses, then mention a line that based on what the doses are prepared.

Response: Thank you. We mentioned this in the method section, but abstract is not a right palce to add such references.

I wish not to write in details that what is being obtained at what particular dose. Just mention the primary results and discuss in 1-2 lines, state the application and future prospective,

Response: Thank you. Please note that it is mostly advised to mention all results in the introduction, as it gives more details to the readers. However, as no other reviewers asked for this, if the editor follows your comment, we will eagerly summarize the results.

Keywords: Ok

Introduction

Line 40-45: Add references when you present data in a line.

Response: Thank you. We added a reference form FAO statistical report.  

48-49: Change : Thermal stress broadly affects physiological processes including oxidative stress physiology in fishes for example, it triggers the stress axis leading to higher energy expenditure and immunosuppression []. It is because temperature can influence mitochodrial respiration and associated events to produce high reactive oxygen species (ROS) in aquatic animals [https://pubmed.ncbi.nlm.nih.gov/24679979/ and some other refernces}.

Response: Thank you very much. We followed your comment.

Line 50-51: It does not link with the previous line, so change it to “It increases the chance of energy expenditure that leads to produce of excess pro-oxidant molecules………..

 Response: Thank you. Considering your previous comment and this one, we changed this par entirely. Please check it.

Line 54-67: Reduce it to 3-5 lines and add the thymol (next para ie line 69-84) para with this.

Response: Thank you. We changed this section, according to your comment.

79-81: I  suspect about the novelty of the work: If the authors declare that such studied are already there, then what is your current hypothesis?? Please clearly mention the hypothesis based on literature especially reference 15.

Response: Thank you. Please note that no study (even reference 15) has been deal with the effects of dietary thymol on heat stress responses in rainbow trout. So this is the first study. Also, regarding the reference 15, this study is strange! The fish growth was very poor and the doses used is very high. I read this reference and followed the link the authors provided for thymol supplied. There was a clear difference between the thymol form the authors claimed and whet presents in the supplier website. So I suppose the authors may diluted the purchased thymol, but they forgot to apply the dilution factor when wrote the article.

Materials and methods

Section 2.1 Based on what you have given this dose. Reference 12 is on craps. You had to determine the doses.

Response: Thank you. As I responded to your previous comment, there is only one study on trout, which was not reliable for us. So we used the doses used in other fish species. The optimum thymol concentration for highest feed efficiency in snakehead was 300 mg/kg (https://doi.org/10.1111/anu.13217), in grass carp was 100 mg/kg (https://doi.org/10.1007/s10695-019-00718-2), in Nile tilapia was 200 mg/kg (https://doi.org/10.1007/s10695-019-00718-2). We used a wide range of dietary thymol even below 100 mg/kg and above 300 mg/kg.

Whether fish were disinfected after collection, if yes, mention how and if no then mention why not??

 Response: Thank you. We used 250 ul/l formalin for 30 min. It was added to the methods.

Section 2.3: Mention why 25 C is thermal stress to this fish.

Response: Thank you. Please note that we used this based on a previous study on the same species (the reference had been added already)

Section 2.5.3: Mention in detail about the composition of the buffer, its pH, antiproatse etc.

Response: Thank you. Please note that “pH” was already mentioned. We added molarity. No other specification such as antiprotease ….

Results

Table 2, 3, figure 1: with  n=3 cant give you a concluding remark on growth rate

Response: Thank you. Please note that n = 3 means “average of three aquaria per treatment”, not three fish per treatment/aquarium. In aquaculture, rearing tank is considered as “replication” not the individuals.

Figure 2-5,  need two way ANOVA test

Response: Thank you. Please note that they are analyzed using two-way ANOVA. Above each graph, you can find the results of two-way ANOVA. When there was an interaction effect, we used Duncan for pair comparison.

Discussion

OK

Response: Thank you.

Conclusion

Ok

Response: Thank you.

Reviewer 4 Report

Comments and Suggestions for Authors

This paper examined growth and humoral innate immunity in rainbow trout fingerlings during eight weeks of thymol supplementation (water temperature 13C), then subsequent antioxidant profiles and mortality during 48 hours of heat stress (water temperature 25C). The introduction succinctly established the significance of heat stress in fish and the benefits of feed supplements such as thymol. The materials and methods section is sufficiently detailed to replicate the study, and the statistical analysis is clearly explained. Results are well-presented in tables and graphs with p-values to three significant figures. The discussion is logical, compared the results obtained with the literature and suggested future studies. Whilst the paper concluded 50-800 mg per kg dietary thymol is not a growth-promoting agent in rainbow trout fingerlings and showed no benefits on humoral immune parameters, it is a powerful antioxidant which provides resistance to thermal stress in rainbow trout.

The main comments are about clarity and definitions

Line 19 and 29 CT. Presumably CT means control. Define all acronyms on first use to remove ambiguity.

Line 94 Table 1 Corn meal CTL. Specify what CTL means.

Line 102 ‘fed the above diets (in triplicate) to apparent satiety for eight weeks.’ Clarify whether table 1 is the base diet formulation and the six experimental diets differ by 0, 50, 100, 200, 400 and 800 mg/kg thymol.

Line 110 Specify whether data (12.7 ± 0.43°C, 7.88 ± 0.26 and 7.84 ± 0.84 mg/L) are mean ± standard deviation. Likewise for line 112, Table 2, Table 3 and Table 4.

Figure 1, 2, 3, 4, 5 – specify whether the bars on the graphs are standard deviation or standard error.

Line 116 (2) specify whether Ln means natural log (which is often denoted with a lower-case l)

Line 120 ‘Water temperature elevation from 13 to 25 °C lasted 12 h’ Specify whether this means the water temperature increased by one degree Celsius per hour. If so, state the minimum increment of temperature increase e.g. was the thermostat increased by 1C every hour or was it increased by equal increments which added to 1C across each hour?

Line 208-210 ‘Fish survival after the thermal stress showed significant difference among the treatments, so that significant elevations were found in survival in fish fed diets containing 40-400 mg/kg thymol’. The lowest dose of dietary thymol is 50 mg/kg so this section should refer to diets containing 50-400 mg/kg thymol.

Comments on the Quality of English Language

Comments/suggestions about English are mainly about correct reference to singular or plural and use of the definite and indefinite article (i.e. the vs a/an). There are some incorrect word choices and confusing sections.

Line 40 ‘Rainbow trout, Oncorhynchus mykiss, is an aquaculture candidate in world, where cold water are available.’ This sentence in incomplete. Suggest writing ‘Rainbow trout (Oncorhynchus mykiss) is a potential aquaculture species where cold water is available.’

Line 44 ’however, rainbow trout culture industry faces various obstacles’ write ‘however, the rainbow trout culture industry faces various obstacles’

Line 45 ‘climate changing and warming’ write ‘climate change and warming’

Line 50 ‘production of excess pro-oxidant molecules in fish body that triggers the antioxidant system’ write ‘production of excess pro-oxidant molecules that trigger the antioxidant system’

Line 54 ‘Functional feeds have gained a great attention in aquaculture industry’ write ‘Functional feeds have gained great attention in the aquaculture industry

Line 56 ‘tolerate stressors that are evitable in aquaculture settings’ suggest writing ‘tolerate stressors in aquaculture settings’.

Line 70 ‘a well-known antioxidant compound has been studied’ write ‘a well-known antioxidant compound that has been studied’

Line 77 ‘performance of the fish in all treatments were poor’ write ‘performance of the fish in all treatments was poor’

Line 81 ‘overall performance of the fish were poor’ write ‘overall performance of the fish was poor’

Line 98 ‘Rainbow trout (~5 g individual weight) was purchased’ write ‘Rainbow trout (~5 g individual weight) were purchased’

Line 99 ‘and transported to laboratory’ write ‘and transported to the laboratory’

Line 114 ‘Total feed intake and biomass of each aquarium was measured’ write ‘Total feed intake and biomass of each aquarium were measured’

Line 120 ‘using submersible heater’ write ‘using a submersible heater’

Line 224 ‘there were no significant difference in the enzyme’ write ‘there were no significant differences in the enzyme’

Line 238 ‘Fish fed diets contacting 100 and 200 mg/kg thymol’ ‘Contacting’ should be ‘containing’.

Line 308 ‘Increase in water temperature triggers stress axis in fish’ write ‘Increase in water temperature triggers the stress axis in fish’

Line 309 ‘These changes provide demanded energy to cope with the negative consequences of the thermal stress’ Change ‘demanded’ to ‘required’

Line 316 ‘showed to mitigate stress responses to water temperature elevation’ write ‘mitigated stress responses to water temperature elevation’

Line 321 ‘both led to increase in catabolism processes’ write ‘both led to increases in catabolic processes’

Line 324 ‘Also, GSH have radical scavenging activity’ write ‘Also, GSH has radical scavenging activity’

Line 329 ‘Supporting these, the antioxidant enzymes showed elevations in activity’ suggest writing ‘Similarly, the antioxidant enzymes showed elevations in activity’

Line 330 ‘an increase in the hepatic MDA concentration suggests there was some levels’ write ‘an increase in the hepatic MDA concentration suggests there were some levels’

Line 334 ‘Beside liver, erythrocytes have a critical dependency to the antioxidant for proper functioning. It is because of the presence of heme protein and constant exposure to oxygen molecules’ suggest writing ‘Besides liver, erythrocytes have a critical dependency on antioxidants for proper functioning due to the presence of heme protein and constant exposure to oxygen molecules’

Line 346 ‘subsequent resistances against bacterial infections’ write ‘subsequent resistance to bacterial infections’

Line 354 Reference 39 is duplicated

Line 354 ‘Moreover, well-functioning antioxidant system in the fish liver’ write ‘Moreover, well-functioning antioxidant systems in the fish liver’

Author Response

This paper examined growth and humoral innate immunity in rainbow trout fingerlings during eight weeks of thymol supplementation (water temperature 13C), then subsequent antioxidant profiles and mortality during 48 hours of heat stress (water temperature 25C). The introduction succinctly established the significance of heat stress in fish and the benefits of feed supplements such as thymol. The materials and methods section is sufficiently detailed to replicate the study, and the statistical analysis is clearly explained. Results are well-presented in tables and graphs with p-values to three significant figures. The discussion is logical, compared the results obtained with the literature and suggested future studies. Whilst the paper concluded 50-800 mg per kg dietary thymol is not a growth-promoting agent in rainbow trout fingerlings and showed no benefits on humoral immune parameters, it is a powerful antioxidant which provides resistance to thermal stress in rainbow trout.

The main comments are about clarity and definitions

Line 19 and 29 CT. Presumably CT means control. Define all acronyms on first use to remove ambiguity.

Response: Thank you. Please check the first live of abstract, where they had been already described.

Line 94 Table 1 Corn meal CTL. Specify what CTL means.

Response: Sorry, this table was totally wrong. We revised it.

Line 102 ‘fed the above diets (in triplicate) to apparent satiety for eight weeks.’ Clarify whether table 1 is the base diet formulation and the six experimental diets differ by 0, 50, 100, 200, 400 and 800 mg/kg thymol.

Response: please check the table 1. We clarified it.

Line 110 Specify whether data (12.7 ± 0.43°C, 7.88 ± 0.26 and 7.84 ± 0.84 mg/L) are mean ± standard deviation. Likewise for line 112, Table 2, Table 3 and Table 4.

Response: Thank you. We clarified them all.

Figure 1, 2, 3, 4, 5 – specify whether the bars on the graphs are standard deviation or standard error.

Response: Thank you. We clarified them all.

Line 116 (2) specify whether Ln means natural log (which is often denoted with a lower-case l)

Response: Thank you. Yes. We changed them to “ln”.

Line 120 ‘Water temperature elevation from 13 to 25 °C lasted 12 h’ Specify whether this means the water temperature increased by one degree Celsius per hour. If so, state the minimum increment of temperature increase e.g. was the thermostat increased by 1C every hour or was it increased by equal increments which added to 1C across each hour?

Response: Thank you. We used a heater equipped with thermostat. It was added to the methods.

Line 208-210 ‘Fish survival after the thermal stress showed significant difference among the treatments, so that significant elevations were found in survival in fish fed diets containing 40-400 mg/kg thymol’. The lowest dose of dietary thymol is 50 mg/kg so this section should refer to diets containing 50-400 mg/kg thymol.

Response: Thank you very much. We revised it.

Comments/suggestions about English are mainly about correct reference to singular or plural and use of the definite and indefinite article (i.e. the vs a/an). There are some incorrect word choices and confusing sections.

Line 40 ‘Rainbow trout, Oncorhynchus mykiss, is an aquaculture candidate in world, where cold water are available.’ This sentence in incomplete. Suggest writing ‘Rainbow trout (Oncorhynchus mykiss) is a potential aquaculture species where cold water is available.’

Response: Thank you. We followed your comment.

Line 44 ’however, rainbow trout culture industry faces various obstacles’ write ‘however, the rainbow trout culture industry faces various obstacles’

Response: Thank you. We followed your comment.

Line 45 ‘climate changing and warming’ write ‘climate change and warming’

Response: Thank you. We followed your comment.

Line 50 ‘production of excess pro-oxidant molecules in fish body that triggers the antioxidant system’ write ‘production of excess pro-oxidant molecules that trigger the antioxidant system’

Response: Thank you. This section was totally rewritten.

Line 54 ‘Functional feeds have gained a great attention in aquaculture industry’ write ‘Functional feeds have gained great attention in the aquaculture industry

Response: Thank you. This section was totally rewritten.

Line 56 ‘tolerate stressors that are evitable in aquaculture settings’ suggest writing ‘tolerate stressors in aquaculture settings’.

Response: Thank you. This section was totally rewritten.

Line 70 ‘a well-known antioxidant compound has been studied’ write ‘a well-known antioxidant compound that has been studied’

Response: Thank you. We followed your comment.

Line 77 ‘performance of the fish in all treatments were poor’ write ‘performance of the fish in all treatments was poor’

Response: Thank you. We followed your comment.

Line 81 ‘overall performance of the fish were poor’ write ‘overall performance of the fish was poor’

Response: Thank you. We followed your comment.

Line 98 ‘Rainbow trout (~5 g individual weight) was purchased’ write ‘Rainbow trout (~5 g individual weight) were purchased’

Response: Thank you. We followed your comment.

Line 99 ‘and transported to laboratory’ write ‘and transported to the laboratory’

Response: Thank you. We followed your comment.

Line 114 ‘Total feed intake and biomass of each aquarium was measured’ write ‘Total feed intake and biomass of each aquarium were measured’

Response: Thank you. We followed your comment.

Line 120 ‘using submersible heater’ write ‘using a submersible heater’

Response: Thank you. We followed your comment.

Line 224 ‘there were no significant difference in the enzyme’ write ‘there were no significant differences in the enzyme’

Response: Thank you. We followed your comment.

Line 238 ‘Fish fed diets contacting 100 and 200 mg/kg thymol’ ‘Contacting’ should be ‘containing’.

Response: Thank you. We followed your comment.

Line 308 ‘Increase in water temperature triggers stress axis in fish’ write ‘Increase in water temperature triggers the stress axis in fish’

Response: Thank you. We followed your comment.

Line 309 ‘These changes provide demanded energy to cope with the negative consequences of the thermal stress’ Change ‘demanded’ to ‘required’

Response: Thank you. We followed your comment.

Line 316 ‘showed to mitigate stress responses to water temperature elevation’ write ‘mitigated stress responses to water temperature elevation’

Response: Thank you. We followed your comment.

Line 321 ‘both led to increase in catabolism processes’ write ‘both led to increases in catabolic processes’

Response: Thank you. We followed your comment.

Line 324 ‘Also, GSH have radical scavenging activity’ write ‘Also, GSH has radical scavenging activity’

Response: Thank you. We followed your comment.

Line 329 ‘Supporting these, the antioxidant enzymes showed elevations in activity’ suggest writing ‘Similarly, the antioxidant enzymes showed elevations in activity’

Response: Thank you. We followed your comment.

Line 330 ‘an increase in the hepatic MDA concentration suggests there was some levels’ write ‘an increase in the hepatic MDA concentration suggests there were some levels’

Response: Thank you. We followed your comment.

Line 334 ‘Beside liver, erythrocytes have a critical dependency to the antioxidant for proper functioning. It is because of the presence of heme protein and constant exposure to oxygen molecules’ suggest writing ‘Besides liver, erythrocytes have a critical dependency on antioxidants for proper functioning due to the presence of heme protein and constant exposure to oxygen molecules’

Response: Thank you. We followed your comment.

Line 346 ‘subsequent resistances against bacterial infections’ write ‘subsequent resistance to bacterial infections’

Response: Thank you. We followed your comment.

Line 354 Reference 39 is duplicated

Response: Thank you. We fixed it.

Line 354 ‘Moreover, well-functioning antioxidant system in the fish liver’ write ‘Moreover, well-functioning antioxidant systems in the fish liver’

Response: Thank you. We followed your comment.

Round 2

Reviewer 3 Report

Comments and Suggestions for Authors

The abstract needs changes as per the previous comments. The authors have replied that to the comments "If above are doses, then mention a line that based on what the doses are prepared." that 

Response: Thank you. We mentioned this in the method section, but abstract is not a right palce to add such references. I wish not to write in details that what is being obtained at what particular dose. 

Again they reply to the comment "Just mention the primary results and discuss in 1-2 lines, state the application and future prospective," that 

Response: Thank you. Please note that it is mostly advised to mention all results in the introduction, as it gives more details to the readers. However, as no other reviewers asked for this, if the editor follows your comment, we will eagerly summarize the results.

It seems the authors have a pre-communication with the rest of the reviewers and the handling editor that what ever they reply, the ms will be accepted. If not, they must tone their writing voice and modify the abstract as per the comments to stand it alone, which is the motto of every scientific Journal. 

Thanks you 

Author Response

The  manuscript animals-323130 describes about the increase in oxidative heath status in rainbow trout with thymol supplementation under thermal stress. Followings are the observations
Simple summary; Please include line what was the background of adding thymol rich diet
Response: Thank you. We added this sentence “Heat stress induces oxidative stress in fish, hence using functional diet supplemented with anti-oxidant additives, such as thymol, can be useful to counteract the impact of global warming on aquaculture.” Please check the L14-16.
Abstract
Line 13-20: what are CT and TM, please remember that abbreviations are mentioned first in full before they are cited in abstract and again in main text.
Response: Thank you. Please note that we described them in the first line of abstract, the L21-22.
What  are  0 (CT), 50 19 (50TM), 100 (100TM), 200 (200TM), 400 (400TM) and 800 (800TM), write 50 or 50TM not both. And write like 50, 100, 200, 400 and 800 TM
Response: Thank you. Please note that, for example, “50TM” refers to a diet containing 50 mg/kg thymol. In fact we used these codes for the diets.
If above are doses, then mention a line that based on what the doses are prepared.
Response: Thank you. We mentioned this in the method section (L83-85), but abstract is not a right place to add such references.
I wish not to write in details that what is being obtained at what particular dose. Just mention the primary results and discuss in 1-2 lines, state the application and future prospective,
Response: Thank you. Please note that it is mostly advised to mention all results in the introduction, as it gives more details to the readers. However, as no other reviewers asked for this, if the editor follows your comment, we will eagerly summarize the results.

Keywords: Ok
 Response: Thank you. 
Introduction
Line 40-45: Add references when you present data in a line.
Response: Thank you. We added a reference form FAO statistical report.  Please check the L47.
48-49: Change : Thermal stress broadly affects physiological processes including oxidative stress physiology in fishes for example, it triggers the stress axis leading to higher energy expenditure and immunosuppression []. It is because temperature can influence mitochodrial respiration and associated events to produce high reactive oxygen species (ROS) in aquatic animals [https://pubmed.ncbi.nlm.nih.gov/24679979/ and some other refernces}.
Response: Thank you very much. We followed your comment. Please check the L50-56.
Line 50-51: It does not link with the previous line, so change it to “It increases the chance of energy expenditure that leads to produce of excess pro-oxidant molecules………..
 Response: Thank you. Considering your previous comment and this one, we changed this par entirely. Please check the L50-56.
Line 54-67: Reduce it to 3-5 lines and add the thymol (next para ie line 69-84) para with this.
Response: Thank you. We changed this section, according to your comment. Please check the L57-63.
79-81: I  suspect about the novelty of the work: If the authors declare that such studied are already there, then what is your current hypothesis?? Please clearly mention the hypothesis based on literature especially reference 15.
Response: Thank you. Please note that no study (even reference 15) has been deal with the effects of dietary thymol on heat stress responses in rainbow trout. So this is the first study. Also, regarding the reference 15, this study is strange! The fish growth was very poor and the doses used is very high. I read this reference and followed the link the authors provided for thymol supplied. There was a clear difference between the thymol form the authors claimed and whet presents in the supplier website. So I suppose the authors may diluted the purchased thymol, but they forgot to apply the dilution factor when wrote the article.

Materials and methods
Section 2.1 Based on what you have given this dose. Reference 12 is on craps. You had to determine the doses.
Response: Thank you. As I responded to your previous comment, there is only one study on trout, which was not reliable for us. So we used the doses used in other fish species. The optimum thymol concentration for highest feed efficiency in snakehead was 300 mg/kg (https://doi.org/10.1111/anu.13217), in grass carp was 100 mg/kg (https://doi.org/10.1007/s10695-019-00718-2), in Nile tilapia was 200 mg/kg (https://doi.org/10.1007/s10695-019-00718-2). We used a wide range of dietary thymol even below 100 mg/kg and above 300 mg/kg.
Whether fish were disinfected after collection, if yes, mention how and if no then mention why not??
 Response: Thank you. We used 250 ul/l formalin for 30 min. It was added to the methods. Please check the L93-94
Section 2.3: Mention why 25 C is thermal stress to this fish.
Response: Thank you. Please note that we used this based on a previous study on the same species (the reference had been added already L119)
Section 2.5.3: Mention in detail about the composition of the buffer, its pH, antiproatse etc.
Response: Thank you. Please note that “pH” was already mentioned. We added molarity. No other specification such as antiprotease …. Please check the L170.
Results
Table 2, 3, figure 1: with  n=3 cant give you a concluding remark on growth rate
Response: Thank you. Please note that n = 3 means “average of three aquaria per treatment”, not three fish per treatment/aquarium. In aquaculture, rearing tank is considered as “replication” not the individuals.
Figure 2-5,  need two way ANOVA test
Response: Thank you. Please note that they were analyzed using two-way ANOVA (please check L192). Above each graph, you can find the results of two-way ANOVA. When there was an interaction effect, we used Duncan for pair comparison.
Discussion
OK
Response: Thank you.
Conclusion
Ok
Response: Thank you.